# Immunogenicity of COVID-19 Vaccinations in Hematological Patients: 6-Month Follow-Up and Evaluation of a 3rd Vaccination

**DOI:** 10.3390/cancers14081962

**Published:** 2022-04-13

**Authors:** Lorenz Schubert, Maximilian Koblischke, Lisa Schneider, Edit Porpaczy, Florian Winkler, Ulrich Jaeger, Stephan Blüml, Helmuth Haslacher, Heinz Burgmann, Judith H. Aberle, Stefan Winkler, Selma Tobudic

**Affiliations:** 1Division of Infectious Diseases and Tropical Medicine, Department of Medicine I, Medical University of Vienna, 1090 Vienna, Austria; lorenz.schubert@meduniwien.ac.at (L.S.); lisa.schneider@meduniwien.ac.at (L.S.); florian.winkler@meduniwien.ac.at (F.W.); heinz.burgmann@meduniwien.ac.at (H.B.); stefan.winkler@meduniwien.ac.at (S.W.); 2Center for Virology, Medical University of Vienna, 1090 Vienna, Austria; maximilian.koblischke@meduniwien.ac.at (M.K.); judith.aberle@meduniwien.ac.at (J.H.A.); 3Division of Hematology and Hemostaseology, Department of Medicine I, Medical University of Vienna, 1090 Vienna, Austria; edit.porpaczy@meduniwien.ac.at (E.P.); ulrich.jaeger@meduniwien.ac.at (U.J.); 4Division of Rheumatology, Department of Medicine III, Medical University of Vienna, 1090 Vienna, Austria; stephan.blueml@meduniwien.ac.at; 5Department of Laboratory Medicine, Medical University of Vienna, 1090 Vienna, Austria; helmuth.haslacher@meduniwien.ac.at

**Keywords:** hematology–general, COVID-19, vaccine, clinical trials

## Abstract

**Simple Summary:**

Here we confirm lower humoral and cellular responses in hematological patients compared with controls and highlight the response in risk-groups such as CAR-T-cell recipients. The main risk factor for a poor humoral response was anti-CD20 therapy. CD19^+^ B-cell and CD4^+^ T-cell levels were shown to be additional predictive markers for seroconversion. Further, we demonstrate a decline in antibodies over six months in responders and controls, and question if second vaccination non-responders benefit from of a third vaccination.

**Abstract:**

Here we analyzed SARS-CoV-2-specific antibodies and T-cell responses after two coronavirus disease 2019 vaccinations over a six-month period in patients with hematological malignancies and assessed the effect of a third vaccination in a subgroup. Sixty-six patients and 66 healthy controls were included. After two vaccinations seroconversion was seen in 52% and a T-cell-specific response in 59% of patients compared with 100% in controls (*p* = 0.001). Risk factors for a poor serological response were age (<65a), history of anti-CD20 therapy within the year preceding vaccination, CD19^+^ B-cells < 110/µL, and CD4^+^ T-cells > 310/µL. The magnitude of T-cell response was higher in patients <65a and with CD19^+^ B-cells < 110/µL. Patients and healthy controls demonstrated a significant decrease in SARS-CoV-2 S antibody levels over the period of six months (*p* < 0.001). A third vaccination demonstrated a strong serological response in patients who had responded to the previous doses (*p* < 0.001). The third vaccination yielded seroconversion in three out of 19 patients in those without serological response. We conclude that both humoral and cellular responses after SARS-CoV-2 immunization are impaired in patients with hematological malignancies. A third vaccination enhanced B-cell response in patients who previously responded to the second vaccination but may be of limited benefit in patients without prior seroconversion.

## 1. Introduction

The coronavirus disease 2019 (COVID-19) pandemic, caused by severe acute respiratory syndrome coronavirus (SARS-CoV-2), poses a major challenge to the health care systems around the world [1]. Mortality in patients with hematological malignancies and COVID-19 were shown to be substantially higher in comparison to the general population [2]. Mortality differed between malignant subtypes. It was highest in patients suffering from acquired bone marrow failure syndromes (53%), followed by acute leukaemias (41%), myeloproliferative neoplasms (34%), plasma cell dyscrasias (33%), and lymphomas (32%) [2]. Given the high case-fatality rate in this cohort, COVID-19 vaccinations may provide an essential tool to reduce mortality. Several vaccine candidates have been developed. However, patients with hematological diseases were excluded from the corresponding clinical trials [3,4].

Concerns regarding the serological response to vaccines in patients with hematological malignancies are based either on the direct impact of the underlying disease or the administered therapy [5,6,7,8,9]. In a recent analysis of 167 adults with chronic lymphatic leukemia, none of the patients under anti-CD-20 treatment, administered within a year before vaccination responded to the vaccination [6]. Although humoral immunity is critical to provide protection against SARS-CoV-2, several reports suggest the contribution of T-cell responses, which could be preserved even in the absence of seroconversion [10,11,12,13].

In addition, hematological patients are at an increased risk of breakthrough infection, especially in terms of accelerated waning of antibody levels over time and increasing immune-evasion capabilities of emergent variants [14,15]. A third vaccination was evaluated in hemato-oncological patients. In a retrospective cohort analysis a third vaccination yielded improved antibody levels, which were higher in patients with solid tumors in comparison to patients with haematological malignancies [16]. In all, effectiveness in hemato-oncological patients appears to be dependent on ongoing chemotherapy [16,17].

In the present study, we aim to evaluate serological and T-cell responses after two doses of mRNA COVID-19 vaccination and describe the efficacy of a third vaccine dose in patients not responding to the first two immunizations.

## 2. Materials and Methods

The study is part of a prospective cohort study performed at the Medical University of Vienna, Vienna, Austria, “Characterization of the responsiveness after mRNA SARS-CoV-2 vaccination in patients with immunodeficiency or immunosuppressive therapy”; Eudra CT Nr. 2021-000291-11. Ethical approval for this study was granted by the local ethics committee of the Medical University of Vienna, Austria (EK Nr. 1073/2021).

Patients with hematologic malignancies older than 18 years, who were treated at the Clinical Division of Hematology and Hemostaseology, were enrolled in the study. Further, 66 age- and sex-matched healthy controls (HCs) were included. All patients and HCs were vaccinated as participants of the Austrian vaccination program. Antibodies against the SARS-CoV-2 receptor-binding domain and the nucleocapsid protein were determined before vaccination as well as 10–20 days after the first, 12–28 days, and 5–6 months after the second vaccination. T-cell responses were evaluated in 32 patients 12–28 days after their second vaccination and in 10 patients 28 days after their third vaccination. At each visit, medication history, disease activity, leukocyte subtypes (below/above lower limit of normal), the possibility of meanwhile SARS-CoV-2-infection (positive antigen- and PCR-tests, symptoms), and vaccination associated adverse events were reported. Figure 1 provides a flowchart comprising cohorts included in the further analyses.

### 2.1. Assessment of Humoral Immunity

Humoral immunity was quantified by assessing antibodies to the receptor-binding domain of the viral spike protein using the Elecsys anti-SARS-CoV-2 S immunoassay [18]. The range of the assay was between 0.4 and 2500.0 BAU/mL. Values >0.8 BAU/mL were considered positive. Results under the lower level of quantification were defined as 0.8 BAU/mL to facilitate calculation. In order to identify previous SARS-CoV-2 infections, nucleocapsid-specific antibodies were determined by using the qualitative Elecsys^®^ Anti-SARS-CoV-2 assay [19]. Antibody tests were performed on a cobas^®^ e801 analyzer (Roche Diagnostics, Rotkreuz, Switzerland) at the Department of Laboratory Medicine, the Medical University of Vienna (certified acc. to ISO 9001:2015 and accredited acc. to ISO 15189:2012).

### 2.2. Assessment of Cellular Immunity

Cellular immunity was assessed by a T-cell stimulation assay. PepMix™ SARS-CoV-2 peptide pools were purchased from JPT (Berlin, Germany). The pools comprised 15-mer peptides overlapping by 11 amino acids (aa) covering the entire sequences of the SARS-CoV-2 spike protein. The spike peptides were split into two sub-pools, S1 (aa 1-643) and S2 (aa 633-1273). Peptides were dissolved in dimethyl sulfoxide and diluted in an AIM-V medium for use in the ELISpot assays. For ex vivo ELISpot assays, PBMCs were thawed. A total of 1–2 × 10^5^ cells per well were incubated with SARS-CoV-2 peptides (2 μg/mL; duplicates), AIM-V medium (negative control; 3–4 wells), or PHA (L4144, Sigma, St. Louis, MO, USA; 0.5 μg/mL; positive control) in 96-well plates coated with 1.5 μg anti-IFN-γ (1-D1K, Mabtech, Stockholm, Sweden) for 24 h. After washing, spots were developed with 0.1 μg biotin-conjugated anti-IFN-γ (7-B6-1, Mabtech), streptavidin-coupled alkaline phosphatase (Mabtech, 1:1000), and 5-bromo-4-chloro-3-indolyl phosphate/nitro blue tetrazolium (Sigma). Spots were counted using a Bio-Sys Bioreader 5000 Pro-S/BR177 (Karben, Germany) and Bioreader software generation 10. T-cell responses were considered positive when mean spot counts were at least threefold higher than the mean spot counts of the unstimulated wells.

### 2.3. Statistical Analysis

The statistical analysis was performed using SPSS Statistics 27 (IBM, Armonk, NY, USA), and figures were produced in the R packages “ggplot2”, “ggpubr”, and “viridis”. According to the distribution, continuous variables are represented as the median with 25% quartile (Q1) and 75% quartile (Q3). Differences in unpaired groups were assessed using the Wilcoxon rank-sum test. For multiple testing, Bonferroni correction was applied when indicated. Categorical variables are represented as numbers and rates in percent. Differences in categorical variables of unpaired groups were compared using Fisher’s exact test. The association of relevant variables with seroconversion was described with univariate and a multivariable logistic regression analysis adjusting for age and sex.

## 3. Results Multivariate Analysis

### 3.1. Study Participants Characteristics

Overall, 66 patients suffering from hematologic malignant diseases, as well as 66 healthy controls (HCs), were included in the study. The median age of our cohort was 62 (IQR, 20; Q1–Q3, 50–69) years, and 26 (39%) were female. The median age of the HCs was 50 (IQR, 18; Q1–Q3, 39–57), and 26 (39%) were female. Patient’s baseline demographics are highlighted in Table 1.

### 3.2. Humoral Response to mRNA SARS-CoV-2 Vaccination

At the baseline, two patients and three healthy controls had detectable antibodies in the nucleocapsid-based chemiluminescence assay. After the first mRNA SARS-CoV-2 vaccination fewer patients (9/43, 20.9%) showed antibody response in comparison to healthy controls (44/47, 93.6%; *p* < 0.001). Additionally, SARS-CoV-2 S antibody levels were significantly lower in patients (0.4 [IQR, 0; Q1–Q3, 0.4–0.4]) compared to HCs (20.5 [IQR, 63.2; Q1–Q3, 8.9–72.1]; *p* < 0.001).

After the second mRNA SARS-CoV-2 vaccination seroconversion was evident in 52% (34/66) of the patients and in 100% (66/66) of the HCs (*p* < 0.001). The hematological malignancies associated with the lowest response were aggressive non-Hodgkin lymphomas (NHL) (1/16, 6%), followed by indolent NHL (10/20, 50%), CLL (10/16, 63%), multiple myeloma (5/6, 83%), and myeloid diseases (5/5, 100%). One patient with Langerhans cell histiocytosis did seroconvert. One patient each with ALL and Hodgkin’s lymphoma did not seroconvert. Several variables such as age < and >65a, sex, CD8^+^ T-cells < 280/µL and CD19^+^ B-cells < 110/µL, anti-CD20 therapy, and chemotherapy were analysed for association with the rate of seroconversion. The presence of Anti-CD20 therapy within the last year (OR = 0.1, 95% Confidence interval [CI] = 0.03–0.33, *p* = < 0.001), the number of CD19^+^ B-cells < 110/µL (OR = 0.13, 95% CI = 0.04–0.47, *p* = 0.002), and the number of CD4^+^ T-cells < 310/µL (OR = 0.1, 95% CI = 0.01–0.86, *p* = 0.036) were associated with lower rates of seroconversion. Age > 65a correlated with an increased rate of seroconversion (OR = 2.79, 95% CI = 1.02–7.64, *p* = 0.046) (Figure 2). Multivariate analyses, adjusted for age and sex, are highlighted in the supplements (Appendix A).

Obinutuzumab was exclusively administered in patients <65a (<65a, 8 of 17, 44.7%; >65a, 0 of 10) and patients <65a suffered numerically more often from aggressive NHL, associated with the poorest seroconversion rate (<65a, 30% 11 of 37; >65a, 5 of 29). The overall response rate in bone marrow transplant recipients was similar to the other patients in this cohort (*p* = 0.1). It was highest in patients who received allogeneic stem cell transplantation (4 of 4, 100%), followed by a patient who received autologous stem cell transplantation (3 of 5, 60%). Six patients received anti-CD19 CAR T-cells, of which only one patient (17%) showed seroconversion after four weeks.

Overall, patients with hematological malignancies showed lower antibody titers after the second vaccination (1.11 [IQR, 531.9; Q1–Q3, 0.4–342]) in comparison to HCs (1456.5 [IQR, 1653; Q1–Q3, 678.25–2340.25]; *p*-value < 0.001). Antibody titers were lower in patients aged <65a (0.4 [IQR, 130.1; Q1–Q3, 0.4–115]; aged >65a, 49.9 [IQR, 2185.6; Q1–Q3, 0.4–2133]; *p* = 0.04), history of anti-CD-20 therapy within the last 12 months (0.4 [IQR, 0.4; Q1–Q3, 0.4–0.8]; no anti-CD20 therapy within the last 12 months, 130.5 [IQR, 2263,6; 0.4–2239]; *p* < 0.001), CD19^+^ B-cells < 110/µL (0.4 [IQR, 0.74; Q1–Q3, 0.4–0.8]; CD19^+^ B-cells > 110/µL, 82.5 [IQR, 2322.4; Q1–Q3, 0.4–2259.5]; *p* < 0.001), and CD4^+^ T-cells < 310/µL (0.4 [IQR, 0; Q1–Q3, 0.4–0.4]; CD4^+^ T-cells > 310/µL, 0.88 [IQR, 186.6; Q1–Q3, 0.4–187]; *p* = 0.02) (Figure 3).

The history of anti-CD-20 therapy within the last 12 months correlated with lower CD19^+^ B-cells (*p* < 0.001). CD4^+^ T-cell levels correlated with the seroconversion rate of the different hematological malignancies (*p* = 0.037). Additionally, CD4^+^ T-cell levels were lowest in patients with aggressive NHL (358 [IQR, 361; Q1–Q3, 259–620]), followed by indolent NHL (462 [IQR, 658; Q1–Q3, 264–922]) and CLL (1341 [IQR, 1279; Q1–Q3, 699.5–1978.5]). CD4^+^ T-cell levels did not correlate with the history of anti-CD20 therapy within the last 12 months (*p* = 0.14).

SARS-CoV-2 S antibody levels were reassessed 202 days (IQR, 66.5; Q1–Q3, 175–241.5) after the second vaccination in 53 patients and 58 HCs. For further analysis, the patients were divided into two groups in dependence of seroconversion four weeks after the second vaccination (second vaccination responders, 2VR; second vaccination non-responders, 2VNR). Figure 4A,B illustrate courses of SARS-CoV-2 S antibody levels in patients over a six month period of time.

SARS-CoV-2 S antibodies of second vaccination responders and HCs significantly decreased over the course of six months after the second vaccination (2VR 4 weeks after vaccination, 318 [IQR, 2388.93; Q1–Q3, 25.57–2414.5], 2VR six months after vaccination, 146 [IQR, 879.68; Q1–Q3, 21.83–901.5], *p* < 0.001; HCs four weeks after vaccination, (1456.5 [IQR, 1653; Q1–Q3, 678.25–2340.25], HCs six months after vaccination, 614 [IQR, 729.5; Q1–Q3, 294.75–1048], *p* < 0.001). Interestingly, 10 of 32 (31%) of the 2VR demonstrated a delayed increase of SARS-CoV-2 S antibodies between the four-week visit and the six-month visit, compared to 7 of 58 (12%) of the HCs. Further, SARS-CoV-2 S antibodies of patients responding to the second vaccination (2VR) did differ significantly compared to HCs six months after the second vaccination (2VR six months after vaccination, 146 [IQR, 879.68; Q1–Q3, 21.83–901.5], HCs six months after the second vaccination, 614 [IQR, 729.5; Q1–Q3, 294.75–1048], *p* = 0.003).

The third vaccination was performed in 33 patients (19 2VNR, 14 2VR) at a median of 176 (IQR, 44; Q1–Q3 150–216.5) days after the second vaccination. The vaccination was performed in 17 patients (51.5%) with BNT162b2, eight patients (224.24%) with mRNA-1273, and in eight patients (24.24%) with AZD1222. Only 3 of the 19 2VNR experienced seroconversion after the third vaccination (16%). The 2VR demonstrated a significant increase in SARS-CoV-2 S antibody level (before 3rd vaccination control 29.95 [IQR 110.34; Q1–Q3, 10.67–121], after 3rd vaccination control 1902.5 [IQR 1459.75; Q1–Q3, 1040.25–2500], *p* = 0.002). Figure 4C highlights SARS-CoV-2 S antibody levels after the third vaccination in 2VR and 2VNR.

### 3.3. Cellular Response to mRNA SARS-CoV-2 Vaccination

SARS-CoV-2-specific T-cell responses were determined in 32 (49%) patients after the second vaccination and 16 HCs to assess if vaccination induced a cellular immune response. Interestingly, 19 out of 32 patients (59%) demonstrated a T-cell response compared to 100% (16/16) of the HCs (*p* = 0.004). A logistic regression model did not find an association for sex, anti-CD20 therapy within the last year, CD8^+^ T-cells < 280/µL or CD19^+^ B-cells < 110/µL, or frequency of T-cell responses (Figure 2). The multivariate analysis, adjusted for age and sex, is highlighted in the supplements (Appendix A). However, the magnitude of the T-cell response was higher in patients with age <65a (median SFCs/10^6^ PBMC, 256 [IQR, 1064; Q1–Q3, 22–1086]) compared to patients >65a (median SFCs/10^6^ PBMC, 0 [IQR, 91.3; Q1–Q3, 0–65], *p* = 0.002), and CD19^+^ B-cells < 110/µL (median SFCs/10^6^ PBMC 330 [IQR, 1007.5; Q1–Q3, 117.5–1125]), compared to patients with CD19^+^ B-cells within the normal range (median SFCs/10^6^ PBMC 22 [IQR, 265; Q1–Q3, 0–147.5], *p* = 0.03) at 4 weeks after the second vaccine dose. After the third dose, T-cell responses were reassessed in 10 of 22 patients. Five of those ten patients had already mounted T-cell-specific responses after the second vaccination, which increased to 7 out of 10 patients after the third vaccination (Figure 5).

### 3.4. Adverse Events

Data on adverse events were systematically recorded in 66 patients after the first and second vaccinations and 22 patients after the third vaccination. Local and systemic reactions after the first, second, and third vaccine doses are highlighted in Figure 6. No worsening of pre-existing symptoms and no serious adverse events were reported during follow-up. We report one breakthrough SARS-CoV-2 infection in HCs eight months after the second vaccination with remarkably high SARS-CoV-2 S antibodies (2500 BAU/mL) at the six-month follow-up. The patient experienced symptomatic COVID-19 with fatigue, loss of taste, sinusitis, and earache.

## 4. Discussion

Mortality in patients with hematological malignancies and COVID-19 was shown to be substantially higher than in the general population [2]. The serological response is crucial for protection against SARS-CoV-2 [13,20]. In this prospective study, we demonstrate that seroconversion was significantly lower in patients with hematological malignancies in comparison to healthy controls and point out risk factors for a worse serological response. Anti-CD20 therapy, low CD19^+^ B-cell, and low CD4^+^ T-cell count were all associated with impaired antibody responses. Hence, we suggest that CD^+^ 19 B-cell and CD4^+^ T-cell levels could be evaluated as predictive markers for low seroconversion rates. Further, we demonstrate that SARS-CoV-2 S IgG levels significantly decreased over a period of six months. The third vaccination showed a good serological response in patients who had responded to the previous doses but appeared to be of limited value in patients who had no prior serological response.

Overall, patients suffering from hematological malignancies displayed a lower rate of seroconversion when compared to HCs, which was lowest in patients suffering from aggressive NHL, indolent NHL, CLL, multiple myeloma, and myeloid diseases. Several studies demonstrated similar results [6,7,8,9,12]. A prospective study of the European Research Initiative on response rates in CLL patients reported a seroconversion rate of 51% compared to 100% in sex- and age-matched healthy controls [6]. Agha et al. demonstrated an even worse serological response in CLL patients (23.1%) in comparison to lymphomas (31.3%) and multiple myeloma patients (43.3%) [7]. In the latter cohort, myeloid malignancies (6%) responded worst [7]. Secondly, we demonstrate that the rate of seroconversion and levels of anti-SARS-CoV-2 S IgG were significantly dependent on ongoing or recent therapy, particularly on anti-CD-20 treatment. Anti-CD20 therapy was demonstrated to impair serological response in CLL patients and patients with chronic inflammatory disease, especially if administered within the last 12 months [6,21,22,23]. Further, Malard et al. suggested CD19^+^ B-cells as predictive markers for anti-SARS-CoV-2 S IgG levels on day 42 after vaccination [12]. Indeed, we confirm the correlation between anti-CD-20 treatment and lower levels of CD19^+^ B-cells, both of which were associated with lower rates of seroconversion as well as anti-SARS-CoV-2 S IgG levels.

In fact, adapted vaccination regimens for patients under anti-CD20 therapy are currently under investigation. For instance, Marchesi et al. suggested that a break of at least three months between BNT162b2 and the last anti-CD20 dose should be maintained, as patients vaccinated within this period did not respond at all [24].

We demonstrated that levels of CD4^+^ T-cells below the lower range of normal (<310/µL) are significantly correlated with anti-SARS-CoV-2 S IgG levels after the second vaccination. CD4^+^ T-cell levels were particularly low in patients with aggressive NHL, the disease with the lowest humoral response. An analysis of T-cell responses in humans with COVID-19 disease and unexposed individuals demonstrated that virus-specific CD4^+^ T-cells are present in 100% of convalescent patients and that CD4^+^ T cell levels correlated with the magnitude of anti-SARS-CoV-2 S IgG [11]. This is explicable by the findings of Painter et al., who demonstrated that first vaccination-induced antigen-specific CD4^+^ T-cell responses guide the adaptive immune response to the second vaccination [25].

Another risk factor for seroconversion we found was age <65a. This is somewhat in contrast to previous literature, but the uneven distribution of obinutuzumab/rituximab therapy and the numerical higher number of aggressive lymphomas in patients aged <65a might contribute to this finding [26].

In addition to the antibody response to mRNA SARS-CoV-2 vaccination, T-cell responses contributed to vaccine efficacy in healthy controls [9,25,27]. Furthermore, improved survival with a greater number of CD8^+^ T-cells was shown [10]. In our analysis, patients experienced significantly less frequent SARS-CoV-2-specific T-cells responses in comparison to HCs. In contrast to serological responses, the SARS-CoV-2-specific T-cells response rate was lower in patients >65a and CD19^+^ B-cells within the normal range. We did not find a gender-associated difference in the SARS-CoV-2-specific T-cells response [12]. Further, our results revealed lower rates of SARS-CoV-2-specific T-cells in comparison to the cohort of Bange et al., who demonstrated detectable SARS-CoV-2-specific T-cell responses in 77% of the hematological patients, albeit after COVID-19 [10].

A special subgroup in our cohort comprises six anti-CD19 CAR T-cells recipients. Only 1 of 6 patients demonstrated seroconversion. Given the mechanism of action with T-cells eliminating CD19-expressing malignant and normal cells, this finding is not surprising [28].

Another aim of this study was to assess long-term responses after mRNA SARS-CoV-2 vaccinations in hematological patients. In both HCs and patients, the visit after six months demonstrated a significant reduction of anti-SARS-CoV-2 S IgG levels. However, patients responding to the second vaccination more often demonstrated a delayed increase of SARS-CoV-2 S antibodies in comparison to HCs.

Previously, the lack of immunogenicity of mRNA SARS-CoV-2 vaccination has led to evaluations of third vaccine immunization in different COVID-19 high-risk cohorts. Recent reports of transplant recipients demonstrated that a third dose of mRNA vaccine had substantially higher immunogenicity compared to the placebo [29,30]. First reports evaluating the third vaccination in hematological patients have demonstrated an increase in neutralizing antibodies after the third vaccination [16,31,32]. However, ongoing immunosuppressive therapy may limit the effect of the third vaccination, leading to a similar response pattern as observed after the first two doses [17]. Our analysis separated patients with a prior vaccination response from patients without a prior vaccination response. Patients primarily responding to the vaccination demonstrated a significant increase of SARS-CoV-2 S antibodies after the third vaccination. Second vaccination non-responders, however, demonstrated seroconversion in only 3 of 19 patients. Although our study was not designed to draw final conclusions on the immunogenicity of a third immunization, the data indicate that serological non-responders may have a limited benefit from a third vaccination in terms of seroconversion. In our cohort T-cell-specific responses were expanded by a third vaccination from 5 of 10 after the second to 7 of 10 after the third vaccination. A similar rise in T-cell-specific response after the third vaccination was detected in a cohort of 45 hematological patients [31]. Whether the induced cellular immune response contributes to anti-SARS-CoV-2 protection remains to be evaluated.

We report several limitations. In this cohort, hematological patients were older than HCs, as HCs consisted mainly of healthcare professionals working at our center. Hence, the demonstrated difference between patients and HCs could be overestimated, as older age was shown to be a major factor for weaker serological responses [26]. Further, our cohort includes patients with different hematological malignancies, which we were not able to assess separately due to the small sample size. However, our results are in line with previous reports demonstrating the weakest response in disease groups in need of B-cell depleting therapy. Only half of the patients were vaccinated a third time. However, recommendations for a third vaccination were unclear at that time of inclusion, as the effectiveness of the vaccination in hematological patients was debated.

## 5. Conclusions

In conclusion, our findings demonstrate that the full immunization schedule was able to induce anti-SARS-CoV-2 S IgG responses in 52% of patients. B-cell depleting therapy was the major factor for a poor serological response. CD^+^ 19 B-cell and CD4^+^ T-cell levels were shown to be additional predictive markers for seroconversion. T-cell responses were reported in 59% of the patients and were higher in patients with low CD19^+^ B-cell counts. Whether the preserved T-cell response in patients with anti-CD20 therapy and low CD19^+^ B cell counts contribute to some degree of protection requires further evaluation. Over a period of six months, vaccine responders and HCs demonstrated a significant decline in antibodies. In those patients, the third vaccination was demonstrated to be effective in booster B-cell response. In vaccine non-responders, a third vaccination yielded seroconversion in a minority of the patients, suggesting a limited benefit from a third vaccination in terms of seroconversion in this group.

## Figures and Tables

**Figure 1 cancers-14-01962-f001:**
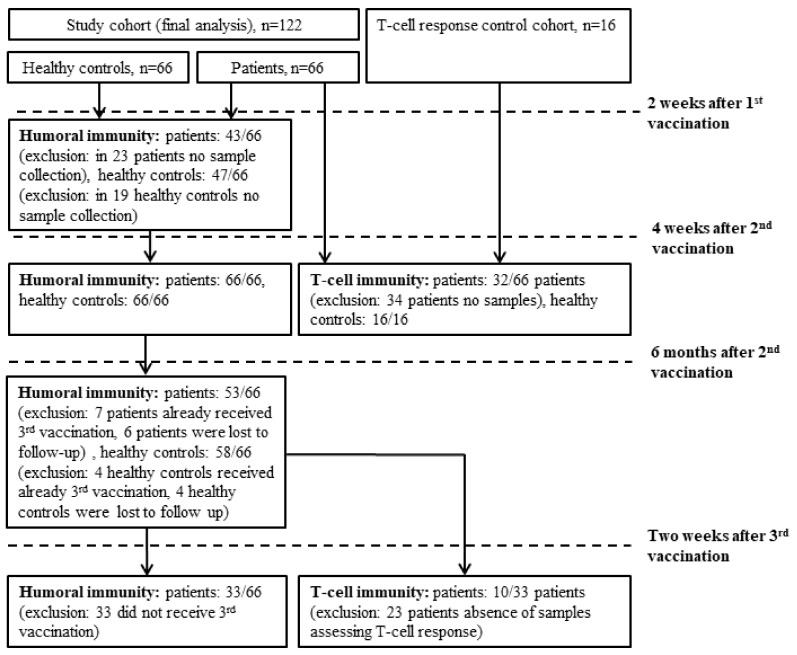
Flow chart demonstrating the cohorts included into the different analysis.

**Figure 2 cancers-14-01962-f002:**
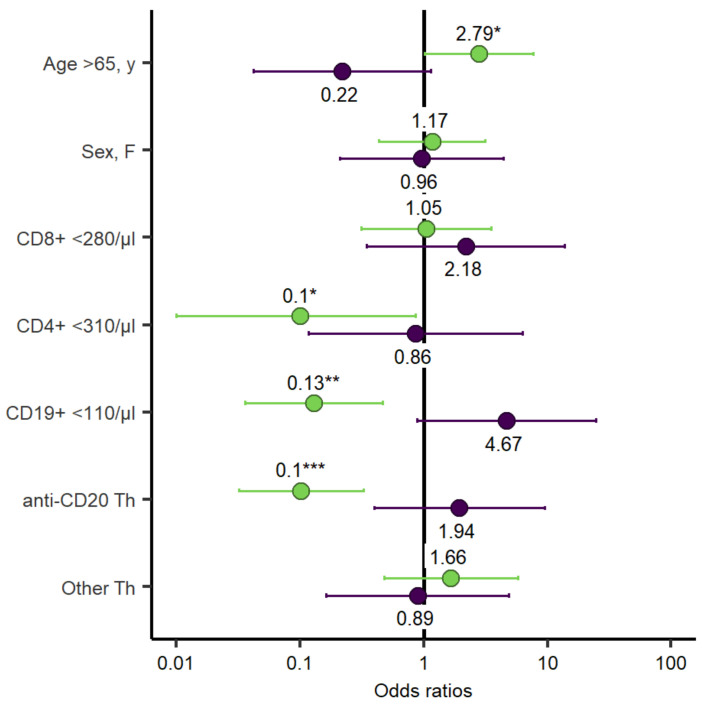
Odds ratio for univariate logistic regression assessing seroconversion and T-cell response. Each of the variables was tested individually against seroconversion (green bars) and T-cell response (purple bars) in all patients. *p*-values are specified as follows: <0.05, *; <0.01, **; <0.001, ***.

**Figure 3 cancers-14-01962-f003:**
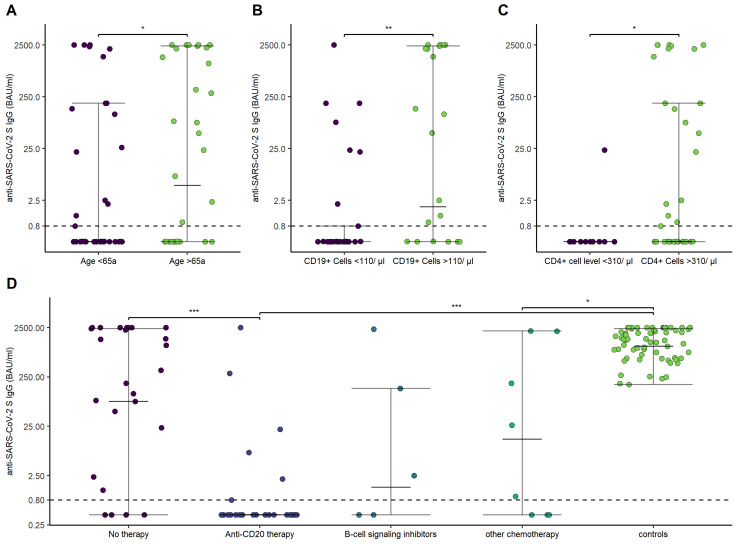
Anti-SARS-CoV-2 S IgG antibodies in relation to age, CD19^+^ B-cells and CD4^+^ T-cells, and different treatments. (**A**) Anti-SARS-CoV-2 S IgG in dependence of age (<65 versus >65a). (**B**) Anti-SARS-CoV-2 S IgG in dependence of CD19^+^ B-cell (<110/µL versus >110/µL). (**C**) Anti-SARS-CoV-2 S IgG in dependence of CD4^+^ T-cell levels (<310/µL versus >310/µL). (**D**) Anti-SARS-CoV-2 S IgG in dependence of treatment and compared to the control group. *p*-values are specified as follows: <0.05, *; <0.01, **; <0.001, ***.

**Figure 4 cancers-14-01962-f004:**
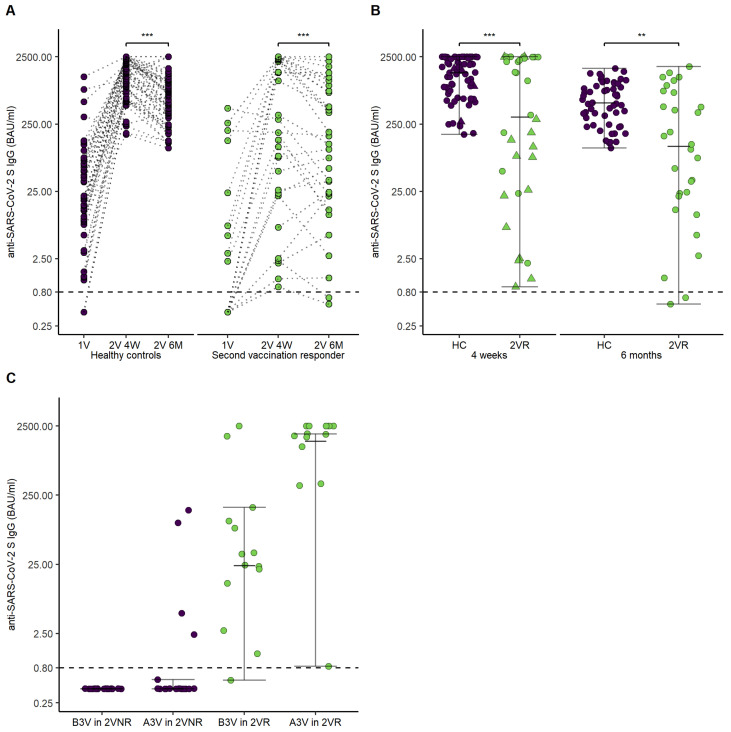
Time course of anti-SARS-CoV-2 S IgG antibodies in healthy controls, second vaccination responders, and second vaccination non-responders. (**A**) Points and lines demonstrate individual anti-SARS-CoV-2 S IgG antibodies’ course after the first vaccination, four weeks after the second vaccination, and six months after the second vaccination for healthy controls and second vaccination responder. (**B**) Error bars demonstrate anti-SARS-CoV-2 S IgG antibodies for healthy controls and second vaccination responder four weeks and six months after the second vaccination. Points demonstrate individual anti-SARS-CoV-2 S IgG antibodies. The triangles show the five patients who received a third vaccination. (**C**) Error bars demonstrate levels of anti-SARS-CoV-2 S IgG antibodies before and after third vaccination in dependence of seroconversion status after the second vaccination. Abbreviations: 1V, first vaccination visit; 2V 4W, second vaccination visit after four weeks; 2V 6M, second vaccination visit after six months; B3V, before third vaccination; A3V, after third vaccination. *p*-values are specified as follows: <0.01, **; <0.001, ***.

**Figure 5 cancers-14-01962-f005:**
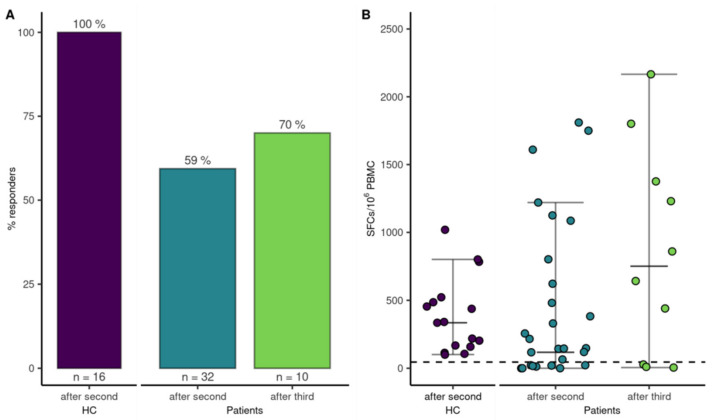
T-cell response to mRNA SARS-CoV-2 vaccination. (**A**) Bars indicate proportion of patients with a T-cell response after the second and third vaccination and HCs after the second vaccination. (**B**) Dots represent quantitative T-cell responses in HCs and patients two-four weeks after the second vaccination and in a subgroup of patients four weeks after the third vaccination.

**Figure 6 cancers-14-01962-f006:**
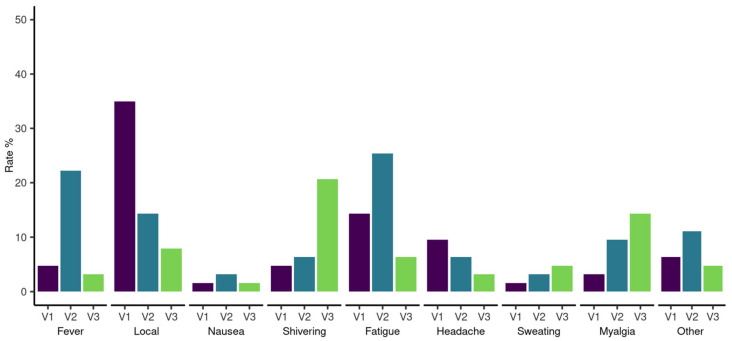
Proportion of adverse events for hematological patients after SARS-CoV-2 vaccination. The proportions are ordered by adverse event and first (V1), second (V2), and third (V3) vaccination.

**Table 1 cancers-14-01962-t001:** Baseline characteristics of the cohort.

Baseline Demographics		Patients (*n* = 66)
Age, median (Q1–Q3), years		62 (50–69)
Females, *n* (%)		26 (39.4)
Male, *n* (%)		40 (60.6)
Hematological malignancies, *n* (%)		
Lymphoid malignancies		
	Chronic lymphocytic leukemia	16 (24.2)
	Indolent NHL	20 (30.3)
	Aggressive NHL	16 (24.2)
	Multiple myeloma	6 (9.1)
	B-lymphoblastic leukemia	1 (1.5)
	Hodgkin lymphoma	1 (1.5)
	Langerhans cell histiocytosis	1 (1.5)
Myeloid malignancies		
	Acute myeloid leukemia	2 (30.3)
	Myeloproliferative neoplasms	3 (45.5)
Disease/treatment status, *n* (%)		
	On-therapy response	35 (53)
	On-therapy partial response	7 (10.6)
	In remission stable	10 (15.2)
	Awaiting therapy	14 (21.2)
Treatment, *n* (%)		
	Anti-CD-20 < 1 year	27 (40.9)
	B-cell signalling inhibitors	5 (7.6)
	Other chemotherapy	8 (12.29)
	No treatment	26 (39.4)
Bone marrow transplantation		
	Autologous stem cell transplantation	5 (7.6)
	Allogenic stem cell transplantation	4 (6.1)
Anti-CD19 CAR T-cells		6 (9.1)
Vaccine, *n* (%)		
	mRNA-1273	35 (53)
	BNT162b2	31 (47)

## Data Availability

We provide access to study documents including the study protocol, blank case report forms as well as anonymized patient-level data after we have fully published all our data on our predefined research objectives. Interested researchers may contact the corresponding author. Data can only be used for scientific research without conflict of interest.

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
