# Peer review of "Immunogenicity of COVID-19 Vaccinations in Hematological Patients: 6-Month Follow-Up and Evaluation of a 3rd Vaccination"

_cancers, 2022, doi:10.3390/cancers14081962_

Round 1

Reviewer 1 Report

I read the manuscript with great interest. The manuscript is well-written and easy to follow.

My minor comments are as below:

  • In the abstract line 26, (HCs) should be added after healthy controls.
  • Line 297: If I understood correctly based on Figure 2c the correlation (positive) should be with >310 and not <310.
  • It would be interesting to add a paragraph on natural course and mortality/morbidity of COVID-19 infection in hematological patients ad compared to healthy individuals (if known).

Reviewer 2 Report

The manuscript is interesting and well written, but some aspects could be improved:

  • Introduction could explain better the background of the study
  • Authors should better point the aim of the study
  • Limitations should be pointed out in the text
  • Inclusion and exclusion criteria of the chosen study population should be explained in the text
  • Some typos are present in the text, a deeper English revision is strongly suggested 

Reviewer 3 Report

The authors present the results of their study on immunogenicity of Covid-19 vaccinations in hematological patients. Several issues have to be addressed:

1) Please provide the criteria of patient and control selection for inclusion in the study. Were the controls matched for some confounding factors with the cases?

2) How the cut-offs for T-cell and B-cell response were selected?

3) Please provide also the results of multivariate analyses regarding the predictive factors for B-cell and T-cell responses following vaccination.

4) Fig. 3 Legend The triangles represent the patients who received a third dose. Please check the numbers for correctness

5) The fact that younger age was associated with reduced antibody production is in contrast to the current literature. The authors should try to investigate the potential confounding factors with appropriate statistical analysis and provide these results.

6) Supplemental Figure should be included as a main figure

7) Please comment on the limited number of included patients in the limitations of the discussion, especially regarding the administration of the third dose and the subgroup analyses on subtypes of hematological malignancies

8) The discussion should be improved with the inclusion of more pertinent references from the literature, since there are plenty of relevant published data. The authors should highlight their unique findings in comparison to the current knowledge

Round 2

Reviewer 3 Report

The authors have addressed the comments